# Effects of Four Strains of *Actinomycetes* on the Content of Terpenoids in Baijiu

**DOI:** 10.3390/foods12071494

**Published:** 2023-04-01

**Authors:** Minxue Feng, Qiaojuan Huo, Linyao Gan, Yefu Chen, Dongguang Xiao, Xuewu Guo

**Affiliations:** 1Key Laboratory of Industrial Fermentation Microbiology of Ministry of Education, Tianjin Industrial Microbiology Key Laboratory, College of Biotechnology, Tianjin University of Science and Technology, Tianjin 300547, China; 2Key Laboratory of Wuliangye-Flavor Liquor Solid-State Fermentation, China National Light Industry, Yibin 644000, China

**Keywords:** actinomycetes, Baijiu, terpenoids, brewing raw materials, fermentation mode, headspace solid-phase microextraction

## Abstract

Terpenoids not only are an important health factor in baijiu but also contribute to the elegance and finesse of baijiu, and actinomycetes act as an important source of terpenoids in baijiu. Four strains of actinomycetes—*Streptomyces violascens* (SPQ1), *S. sampsonii* (SPS1), *S. thermophilus* (SPG1), and *S. griseus* (SPH1)—obtained from the Daqu, pit mud, fermented grains and air, respectively, in the production of baijiu were used in solid-state and liquid fermentation with five brewing raw materials as the substrates. The terpenoids in the metabolites were analyzed and compared using gas chromatography-mass spectrometry (GC-MS). We found that the four strains of actinomycetes produced 31 terpenoids from the hydrolysates of five fermentation substrates during liquid fermentation, and the total terpenoid content was 989.94 μg/kg in the fermentation products. After 28 days of solid-state fermentation, the four actinomycete strains produced 64 terpenoids using the five fermentation substrates, and the total terpenoid content was 23,651.52 μg/kg in the fermentation products. The different fermentation substrates and fermentation methods have a great influence on the terpenoids produced by actinomycetes.

## 1. Introduction

Baijiu is a kind of traditional distilled liquor in China. It is considered to be the most famous distilled liquor in the world together with brandy, gin, rum, vodka and whisky [1]. Baijiu is usually brewed from sorghum alone, or mixed with corn, rice, wheat, peas, millet and sorghum. Cereals are mixed with distiller’s yeast, which is called the base of Baijiu, to produce ethanol and flavor compounds after saccharification and fermentation. Distill the fermented mixture under solid state conditions to produce fresh Baijiu. The body of Baijiu is mainly composed of a water alcohol solution, accounting for 98% (*v*/*v*), but the remaining trace components occupy an absolute position in the dominant flavor value of Baijiu. The flavor type and the style of Chinese Baijiu are both closely related to the 1~2% trace components, which determine the taste and the quality of Baijiu and have a decisive impact on the liquor body. The 12 flavors of Chinese Baijiu are mainly caused by those 1~2% trace components [2,3]. It is reported that there are over 2020 trace components in Baijiu, including esters, acids, terpenes, ester peptides, pyrazines and so on [4].

Terpenoids have been gradually used in the food and beverage industry [5]. They are not only trace components for aroma and flavor in the baijiu body [6], but also healthy bioactive components that have received much attention from researchers in recent years [7]. Terpenoids are a class of naturally occurring unsaturated hydrocarbons derived from the polymerization of two or more isoprene molecules [8,9]. Terpenoids have significant physiological activity, with anticancer, antiviral and anti-inflammatory active properties [10,11,12]. Studies have reported high levels of terpenoids in qingxiangxing baijiu [13,14], jiangxiangxing baijiu [15,16], nongxiangxing baijiu [17] and dongxiangxing baijiu [18,19]. Terpenoids play a role in improving the flavor of baijiu, and some studies have shown that terpenoids are an important aroma component of Moutai flavor baijiu, making it more elegant and delicate [15,20]. Terpenoids are generally thought to come from the following sources: First, they are introduced into the baijiu by the raw materials used in the brewing process. The raw materials used in making baijiu themselves contain some terpenoids that are carried into the baijiu through the brewing process [21]. Second, they are produced by microbial metabolism. In the brewing process, there are thousands of microorganisms, some of which can synthesize terpenoids from scratch or metabolize terpenoids from precursors in raw and auxiliary materials [22,23]. Third, they are generated by precursors through chemical reactions. In the baijiu fermentation process, especially in high-temperature distillation and Jiuqu making, the precursor substances generate terpenoids through chemical reactions.

Actinomycetes are important microorganisms in baijiu brewing. They produce not only important baijiu flavor substances, such as esters, acids, aldehydes, ketones, and alcohols, but also some unique metabolites, such as terpenoids, which have an important impact on baijiu flavor and quality. Du et al. [24] reported that terpenoids in qingxiangxing baijiu are mainly derived from the metabolism of actinomycetes. The actinomycete NS01, which was isolated by Yang Fan from the soil of the jiangxiangxing baijiu production environment, metabolized 17 terpenoids in a 3% flour medium. Twenty terpenoids were metabolized in a solid mixed medium of wheat and sorghum [25].

Although there are some reports of terpenoid production by actinomycetes, there is a lack of systematic and in-depth research on actinomycetes and their use of terpenoids produced by different brewing materials. In this study, four species of actinomycetes from the Daqu, cellar mud, fermented grains and air were used as the research object. After enrichment and cultivation, solid-state fermentation and liquid fermentation were carried out using five traditional baijiu brewing raw materials, namely sorghum, wheat, maize, rice and glutinous rice, as substrates. The terpenoids in the metabolites were analyzed and compared using gas chromatography-mass spectrometry (GC-MS), and the production of terpenoids by the actinomycetes and their characteristics when using different raw materials and different fermentation methods to produce terpenoids were investigated. This study provides a reference value for investigating the sources of terpenoids in baijiu and regulating the terpenoid content of baijiu.

## 2. Materials and Methods

### 2.1. Microorganisms and Media

*S. violascens* (SPQ1) was isolated from the Daqu of nongxiangxing baijiu, *S. sampsonii* (SPS1) was isolated from the pit mud of nongxiangxing baijiu, *S. thermophilus* (SPG1) was isolated from the fermented grains of jiangxiangxing baijiu and *S. griseus* (SPH1) was isolated from the air of the qingxiangxing baijiu brewing environment.

All the *Streptomyces* were inoculated in 250 mL Erlenmeyer flasks with 50 mL of a ISP2 medium (yeast extract 0.4%, malt extract 1%, glucose 0.4%, pH 7.2, and autoclaved at 121 °C for 15 min). Fermentations were conducted at 180 rpm and 30 °C for 48 h.

### 2.2. Fermentation Conditions

#### 2.2.1. Solid-State Fermentation Conditions

Referring to the general scheme for traditional processing in Baijiu production [26], this traditional process was simulated in the laboratory.

The solid-state fermentation medium was composed of crushed brewing raw materials. Sorghum, wheat, corn, rice, glutinous rice and rice husks were all sourced from a winery in Anhui. Three kilograms of sorghum was combined with 2% rice husk and 80% distilled water, and the mixture was used to fill 500 mL Erlenmeyer flasks of 200 g each. Finally, the solid-state fermentation medium was autoclaved at 121 °C for 30 min. After sterilization, the mixture in the bottles was immediately dispersed under sterile conditions. This was the sorghum solid medium to be inoculated. The same operation method was used for the production of wheat, corn, rice and glutinous rice solid media.

The seed cultures of the *Streptomyces* were inoculated into different solid-state fermentation media at a ratio of 10% (*v*/*w*), and the mixture was stirred evenly. The fermentation condition was 30 °C for 28 days. After the solid-state fermentation, the terpenoid compounds of the resulting samples were detected.

#### 2.2.2. Liquid-State Fermentation Conditions

Referring to the general scheme of the traditional process in the liquid-state production of Baijiu, this traditional process was simulated in the laboratory [27].

An amount of 250 g of sorghum was combined with 1 L of water, whereupon a liquefying enzyme (1 mL/kg) was added to the mixture. The mixture was then treated in a 90 °C water bath for 1 h and was boiled for 0.5 h. Next, the temperature was dropped to 60 °C, glucoamylase (2 mL/kg) was added to the mixture, saccharification was performed at 60 °C for 2 h, followed by the addition of acid protease (1 mL/kg) and then the mixture was further treated in a water bath for 2 h. After that, the liquid was cooled and the supernatant was separated. Then the sugar concentration was adjusted to 12°Bx. The resulting mixture was transferred into 250 mL Erlenmeyer flasks of 50 mL each. Finally, the liquid-state fermentation medium was autoclaved at 121 °C for 30 min. This was the sorghum liquid medium to be inoculated. The same operation method was used for the production of wheat, corn, rice and glutinous rice liquid media.

The seed cultures of the *Streptomyces* were inoculated into different liquid-state fermentation media at a ratio of 10% (*v*/*v*). Fermentations were conducted at 180 rpm and 30 °C for 3 days. After the liquid fermentation, the terpenoid compounds of the resulting samples were detected.

### 2.3. Semi-Quantitative Analysis of Terpenoids Using Headspace Solid-Phase Microextraction Coupled with Gas Chromatography-Mass Spectrometry (HS-SPME-GC–MS)

Headspace solid-phase microextraction coupled with gas chromatography-mass spectrometry (HS-SPME-GC–MS) was used for the semi-quantitative analysis of terpenoids in the fermentation sample or fermentation broth. The volatile compounds were determined by comparing the mass spectrum and retention time with the pure standard. The mass spectra of unknown compounds were compared with those in the NIST. The compounds depicted in this study were semi-quantified according to the method described previously [28]. Menthyl acetate was used as an internal standard for quantification; single standard-stock solutions of these volatile compounds were prepared in absolute ethanol.

#### 2.3.1. Solid-State Fermentation Sample Processing

A 1 g fermentation sample was spiked with 8 μL of the internal standards (menthyl acetate, 10 mg/L), saturated with 5 mL of saturated NaCl solution in a 20-mL headspace bottle and sealed with a PTFE-silicone cap. A 50/30 μm DVB/CAR/PDMS fiber (Supelco, Inc., Bellefonte, PA, USA) was used for the extraction.

#### 2.3.2. Liquid-State Fermentation Sample Processing

5 mL of fermentation broth was spiked with 8 μL of the internal standards (menthyl acetate, 10 mg/L), saturated with 2 g of NaCl in a 20 mL headspace bottle and sealed with a PTFE-silicone cap. A 50/30 μm DVB/CAR/PDMS fiber (Supelco, Inc., Bellefonte, PA, USA) was used for the extraction.

#### 2.3.3. HS-SPME Conditions

The headspace bottle with fermentation sample or fermentation broth was equilibrated at 70 °C for 5 min, and the SPME fiber (50/30 μm CAR/DVB/PDMS) was exposed in the headspace bottle at 70 °C for 60 min.

#### 2.3.4. GC-MS Conditions

Thermo Trace 1300 (Thermo Fisher Scientific, Bremen, Germany) equipped with an ISQ Single MS spectrometer (Thermo Fisher Scientific, Austin, TX, USA) was used to analyze the target compound. The samples were injected using a Triplus RSH automatic injector (Thermo Fisher Scientific, Austin, TX, USA). The SPME fiber was placed at the GC injection port and desorbed at 250 °C for 4 min in a splitless mode. Helium (purity > 99.999%) was used as the carrier gas at a flow rate of 1 mL/min. The compounds were separated on an HP-5MS capillary column (30 m × 0.25 mm × 0.25 μm, Agilent Technologies, Inc., Santa Clara, CA, USA). The temperature program was 40 °C for 1 min and was increased to 200 °C at 4 °C/min, followed by an increase at 20 °C/min to 250 °C, which was maintained for 10 min. The temperature of the transfer line and ion source was 250 °C, and the temperature of the quadrupole was 150 °C. The MS operated in electron ionization mode at 70 eV. The mass scan range was 50–400 amu.

### 2.4. Data Analysis

Microsoft Excel was used for statistical analysis and table production of the data; the BioCloud platform (https://www.omicstudio.cn/tool, accessed on 30 March 2023) of Hangzhou Lianchuan Biotechnology Co from Hangzhou, China.

## 3. Results

### 3.1. Analysis of the Liquid Fermentation Terpenoid Metabolites

The fermentation products were extracted using the headspace solid-phase microextraction (HS-SPME) method and then analyzed using GC-MS. The four strains of *Streptomyces* were inoculated into the liquid medium with different substrates at a ratio of 10%, and the corresponding blank medium without inoculated bacteria was used as the control.

From Table 1, the five liquid blank media contain six terpenoids, namely fitone (6.27 μg/kg), linalool (2.13 μg/kg), geranylacetone (1.39 μg/kg), nerolidol (1.79 μg/kg), dihydro-β-ionone (1.45 μg/kg) and α-acorenol (1.40 μg/kg). These terpenoids were derived from the raw materials, but they are not representative of the terpenoids present in the raw materials themselves. Since these raw materials underwent a series of pretreatments and were autoclaved, the terpenoids may have been lost or converted.

The four Streptomyces strains produced 31 terpenoids during liquid fermentation using five hydrolysates as substrates, yielding a total content of 989.94 μg/kg. The terpenoid products of the liquid fermentation were analyzed using the substrate and the strains as the study objects, and the results are given below.

#### 3.1.1. Comparative Analysis of the Products of Different Substrates

The total content of terpenoids produced by different liquid fermentation substrates is shown in Figure 1. When the fermentation substrates were all sorghum hydrolysates, the four *Streptomyces* strains produced 14 terpenoids at 521.56 μg/kg. The highest concentrations of β-eudesmol and fitone were found among the five media, while fitone was detected in the blank sorghum liquid medium. (+)-β-Cedrene, (−)-epiglobulol, β-elemene, zingiberene and β-sesquiphellandrene were only found in the fermentation products when a sorghum liquid medium was used for fermentation.

When the fermentation substrate was wheat hydrolysates, the four strains of *Streptomyces* produced 10 terpenoids at 106.95 μg/kg. Geranyl isovalerate and (−)-epicedrol had the highest concentrations among the five mediums, and hinokitiol and cedrol were only found in the fermentation products when a wheat liquid medium was used for fermentation.

When the fermentation substrate was corn hydrolysates, the four strains of *Streptomyces* produced 14 terpenoids at 95.81 μg/kg. The concentrations of geranylacetone, nerolidol and nerol were the highest among the five media, while dihydro-β-ionone, α-acorenol, isocalamenediol, eremophilene, α-cadinol and β-ionone were only found in the fermentation products when fermented with corn liquid medium. Nerolidol, dihydro-β-ionone and α-acorenol were detected in the blank corn liquid medium.

When the fermentation substrate was rice hydrolysates, the four strains of *Streptomyces* produced seven terpenoids at 73.28 μg/kg. (−)-α-Terpineol was only found in the fermentation product when a rice liquid medium was used for fermentation.

When the fermentation substrate was sticky rice hydrolysates, the four strains of *Streptomyces* produced 14 terpenoids at 192.34 μg/kg. The concentrations of 1-octen-3-ol, (E)-2-octenal, 2,4-decadienal, 2-undecenal, linalool and geraniol were highest in the five media. (E)-2-Heptenal, farnesol, terpinen-4-ol and citronellol were only found in the fermentation products when fermented with sticky rice liquid medium, and linalool was detected in the blank sticky rice liquid medium.

Terpene compounds produced by different grains are sorghum (521.56 μg/kg) > glutinous rice (192.345 μg/kg) > wheat (106.95 μg/kg) > corn (95.81 μg/kg) > rice (73.28 μg/kg), and the amount of terpene compounds produced is sorghum (14 kinds) = corn (14 kinds) = glutinous rice (14 kinds) > wheat (10 kinds) > rice (7 kinds). SPQ1 has the highest ability to produce terpene compounds when sorghum, rice and glutinous rice are used, and SPG1 has the highest ability to produce terpene compounds when wheat and corn are used.

#### 3.1.2. Comparative Analysis of the Products of Different Strains

The total terpenoid content produced by different *Streptomyces* species is shown in Figure 2 and Figure 3. When fermentation was carried out using SPS1, 18 terpenoids were produced at 258.02 μg/kg, yielding the highest levels of geraniol, nerolidol, (+)-β-cedrene and α-acorenol, with terpinen-4-ol, citronellol and cedrol being the special products of SPS1. When SPQ1 was used for fermentation, 16 terpenoids were produced at 423.67 μg/kg, yielding the highest levels of linalool, 1-octen-3-ol, (E)-2-octenal, geranylacetone, β-eudesmol, 2,4-decadienal and 2-undecenal, with (−)-epiglobulol, β-ionone and farnesol as the specific products of SPQ1. When SPG1 was used for fermentation, 21 terpenoids were produced at 148.11 μg/kg, yielding the highest levels of (−)-epicedrol, nerol, hinokitiol, geranyl isovalerate and dihydro-β-ionone, with isocalamenediol, eremophilene, (−)-α-terpineol, zingiberene, β-sesquiphellandrene and α-cadinol as the specific products of SPG1. When SPH1 was used for the fermentation, 15 terpenoids were produced with a terpenoid content of 160.14 μg/kg, with the highest concentrations of fitone, (E)-2-heptenal and β-elemene being the special products of SPH1.

The concentrations of terpene compounds produced by the four strains were SPQ1 (423.67 μg/kg) > SPS1 (258.02 μg/kg) > SPH1 (160.14 μg/kg) > SPG1 (148.11 μg/kg), and the quantities of terpene compounds produced were SPG1 (21 species) > SPS1 (18 species) > SPQ1 (16 species) > SPH1 (14 species). SPS1, SPQ1 and SPH1 had the best effect on terpene production when sorghum liquid fermentation medium was used; SPG1 has the best effect in producing terpene using corn liquid fermentation medium.

### 3.2. Analysis of the Solid-State Fermentation Terpenoid Metabolites

The four strains of *Streptomyces* were each added at 10% to the different solid media of the substrate. The corresponding blank medium without inoculation was used as a control and samples were taken at intervals of 7 days until the end of 28 days of fermentation. The fermentation products were extracted using the headspace solid-phase microextraction (HS-SPME) method and then analyzed using GC-MS.

From Table 2, the solid blank medium contains five terpenoids, namely (E)-2-octenal (0.63 μg/kg), nerolidol (4.62 μg/kg), fitone (28.99 μg/kg), linalool (0.77 μg/kg) and geranylacetone (1.25 μg/kg). It should also be considered that these terpenoids came from the raw materials, but that some pretreatments may have resulted in some loss and conversion of the terpenoids. This also explains the difference in terpenoids between solid and liquid blank fermentation media.

When the solid-state fermentation reached 28 days, the four *Streptomyces* strains used five fermentation substrates to produce 64 types of terpenoids, and the total content of terpenoids reached 23,651.52 μg/kg. The terpenoid products of the solid-state fermentation were analyzed using the substrates and strains as the objects of study, and the results were analyzed as follows.

#### 3.2.1. Comparative Analysis of the Products of Different Substrates

The terpenoids produced from the different substrates when the fermentation reached 28 days are shown in Table 3.

When the fermentation substrate was all sorghum, a total of 38 terpenoids were present during fermentation: (−)-globulol, α-agarofuran, β-caryophyllene, (−)-humulene epoxide II, α-farnesene, (E)-β-famesene, α-gurjenene, guaiol and α-cubebene only appeared in the fermentation process when using sorghum solid medium. On day 28 of fermentation, the terpenoid content reached 9815.65 μg/kg. At this time, SPG1 produced the most terpenoids (18 species) using sorghum solid medium, and the terpenoid content produced by SPS1 reached 7418.96 μg/kg.

When the fermentation substrate was all wheat, a total of 34 terpenoids were present during the fermentation process, with farnesol, β-germacrene, drimenol, viridiflorol and α-pinene only found in the fermentation products when using a wheat solid medium. When the fermentation proceeded to 28 days, the terpenoid content reached 2173.58 μg/kg, at which point SPG1 produced 12 terpenoids using a wheat solid medium, and the terpenoid content produced by SPS1 reached 940.25 μg/kg.

When the fermentation substrate was all corn, a total of 33 terpenoids were present during the fermentation: hinokitiol, 7-epi-sesquithujene, α-curcumene, (E)-2-heptenal, citronellol and 2,6-dimethyl-5-heptenal were only found in the fermentation products when the fermentation was carried out using corn solid medium. When fermentation proceeded to 28 days, the terpenoid content was 8022.28 μg/kg and 11 terpenoids were produced by both SPS1 and SPG1. However, the terpenoid-producing content of SPS1 was much higher than that of SPG1 (6174.71 μg/kg and 360.24 μg/kg).

When the fermentation substrate was all rice, 33 terpenoids appeared during the fermentation process. β-bisabolene, β-sesquiphellandrene, β-selinene, aromandendrene, α-terpieol and α-cadinene were only found in the fermentation products when rice solid medium was used for the fermentation. On the 28th day of fermentation, the terpenoid content reached 2008.42 μg/kg, and both SPS1 and SPQ1 produced 14 terpenoids, with SPH1 producing slightly more terpenoids than SPS1.

When the fermentation substrate was all sticky rice, a total of 27 terpenoids appeared during the fermentation: (E)-2-octen-1-ol and (E)-2-decenal were only found in the fermentation products when using sticky rice solid medium. When the fermentation proceeded to 28 days, the terpenoid content reached 1641.66 μg/kg, at which time SPS1 produced 12 terpenoids and had the highest terpenoid production of the four strains at 563.74 μg/kg.

Figure 4 shows that, until the end of fermentation on day 28, the terpenoids produced by the actinomycetes using sorghum were higher in content and variety than that produced by the other four raw materials. This also confirms the status of sorghum as the main raw material for the brewing of baijiu. The differences in the effects of various materials on terpenoid formation may be due to the different chemical compositions of the raw materials. Therefore, the use of a wider variety of raw materials in the brewing process could provide a broader range of terpenoids and subsequent flavors for baijiu.

Terpene compounds produced by different grains are sorghum (9815.65 μg/kg) > corn (8022.28 μg/kg) > wheat (2173.58 μg/kg) > rice (2008.42 μg/kg) > glutinous rice (1641.66 μg/kg), and the amount of terpene compounds produced is sorghum (35 species) > rice (31 species) > wheat (25 species) > corn (23 species) > glutinous rice (19 species). When sorghum, wheat, corn and glutinous rice are used, SPS1 has the highest ability to produce terpene compounds, and when rice is used, SPH1 has the highest ability to produce terpene compounds.

#### 3.2.2. Comparative Analysis of the Products of Different Strains

The total terpenoid production of different *Streptomyces* is shown in Table 4.

From Figure 5 and Figure 6, when SPS1 was used for fermentation, a total of 24 terpenoids were produced. β-Bisabolene and β-sesquiphellandrene only appeared in the process of fermentation by SPS1. On day 28 of fermentation, SPS1 produced a total of 15,605.74 μg/kg of terpenoids using five raw materials. The largest variety of terpenoids was produced from corn, and the greatest concentrations of terpenoids were produced from sorghum.

When SPQ1 was used for fermentation, a total of 30 terpenoids appeared in the fermentation process. 2-(E)-octen-1-ol, (E)-2-decenal, (−)-globulol and hinokitiol only appeared in the fermentation process when using SPQ1. On day 28 of fermentation, a total of 4337.66 μg/kg terpenoids were produced by SPQ1 from five materials; rice was used to produce the greatest variety of terpenoids, and sorghum was used to produce the greatest concentration of terpenoids.

When SPG1 was used for fermentation, a total of 42 terpenoids appeared in the fermentation process. (−)-Humulene epoxide II, aromandendrene, α-agarofuran, 2-undecenal, β-caryophyllene, farnesal, α-farnesene, cembrene, (E)-β-famesene, trans-farnesol, farnesol, β-germacrene, drimenol, 2,4-decadienal, 7-epi-sesquithujene, α-curcumene, (E)-2-heptenal, citronellol and 2,6-dimethyl-5-heptenal only appeared in the process of SPG1 fermentation. On day 28 of fermentation, a total of 1782.25 μg/kg terpenoids were produced by SPG1 from the raw materials. The greatest variety of terpenoids was produced using sorghum and the greatest concentration of terpenoids was produced using rice.

When SPH1 was used for fermentation, 32 terpenoids appeared in the fermentation process. Guaiol, terpinen-4-ol, α-terpieol, curcumol, viridiflorol, α-pinene, α-cadinene, α-gurjenene and α-cubebene only appeared in the process of SPH1 fermentation. On day 28 of fermentation, the concentration of terpenoids produced by SPH1 from the five materials reached 1935.94 μg/kg. Although the samples cultured with sorghum formed the greatest variety of terpenoids, rice did yield the greatest concentration of terpenoids.

The solid-state blank medium contains a total of five terpenoids; at the end of 28 days of fermentation, the four strains of actinomycetes produced a total of 64 terpenoids. This study confirms the contribution of actinomycetes in the production of terpenoids in the baijiu-making process. While SPQ1 produced far more terpenoids than the other three strains, SPG1 was the best performer in terms of the type of terpenoids produced.

The total concentration of terpene compounds produced by the four strains at 28 days was SPS1 (15,605.74 μg/kg) > SPQ1 (4337.66 μg/kg) > SPH1 (1935.94 μg/kg) > SPG1 (1782.25 μ g/kg), and the number of terpene compounds produced was SPG1 (33 species) > SPH1 (25 species) > SPQ1 (23 species) > SPS1 (22 species). SPS1 and SPQ1 produced the highest concentration of total terpenes when using sorghum solid fermentation medium, while SPG1 produced the highest concentration of total terpenes when using rice solid fermentation medium, and SPH1 produced the highest concentration of total terpenes when using wheat liquid fermentation medium.

## 4. Discussion

During solid-state fermentation, we found that as fermentation time proceeded, terpenoids also changed all the time, and these changes occurred for two reasons. First, over time, the process of fermentation was accompanied with volatilization, and compounds that had been produced and were no longer produced through fermentation decreased in concentration, while compounds that were already in low concentrations directly volatilized and disappeared. Second, some terpenoids were used as, or converted into, precursors or intermediates. Studies have shown that *Saccharomyces cerevisiae* can biotransform geraniol to citronellol under brewing conditions. The biotransformation of monoterpene linalool, α-terpinol, nerolidol and geraniol by *Saccharomyces cerevisiae* was also demonstrated in the model fermentation [29]. It is also possible to speculate on the cause of some of the material changes in our experiments, but the exact mechanism for this is not yet clear.

During the solid-state fermentation of baijiu, a small number of substrates undergo specific carbon atom migration reactions and skeletal rearrangements in the presence of appropriate microorganisms, forming terpenoids through microbial transformation. This approach is an important reason why various aromatic baijiu, such as nongxiangxing and jiangxiangxing, contain terpenoids; microbial transformation is also an important method of synthesizing terpenoids. Variations in the structure and chemical composition of raw materials lead to a diversity in terpenoid profiles. The metabolites of terpenoids produced by microorganisms using different substrates vary considerably, and with different fermentation methods, the growth status of microorganisms and the types of flavor compounds are also very different [30]. Wu et al. [31] used different yeast species to produce terpenoids from different cereal and legume materials through different pathways. It was found that *S. cerevisiae* produced the highest concentrations of terpenoids when fermented with barley extracts, and the total concentration of terpenoids produced when fermenting with pea extracts was slightly lower than the concentrations of terpenoids produced when using barley and sorghum. Fermentation with sorghum extract produced low concentrations of terpenoids but was most prominent when geranylacetone and α-bisabolol were produced.

The terpenoid production from liquid and solid-state fermentations were compared: the solid-state fermentation outperformed the liquid-state fermentation in terms of both the total concentration of terpenoids produced and the variety of terpenoids produced when using these four strains of *Streptomyces.* The solid-state fermentation technology is usually used as the main fermentation method for baijiu to produce the most well-known baijiu drinks with different flavors and characteristics [32]. Studies have shown that solid-state fermentation techniques can easily provide acceptable flavor and taste for baijiu, while liquid fermentation still suffers from poor flavor complexes [33]. Our experiments also verify this and confirm the contribution of actinomycetes in the production of terpenoids during the brewing of baijiu. In addition, the pattern of terpenoid production by actinomycetes was also discovered, providing a direction for the better use of actinomycetes in the brewing process of baijiu.

Our study of the fermentation products revealed that all three strains produced β-eudesmol to varying degrees, except when SPG1 was used in liquid fermentation. β-Eudesmol is a sesquiterpene commonly found in some plants, especially in *Atractylodes lancea* [26] and eucalyptus [34]. β-eudesmol was also detected in the volatile oil of *Anaxagorea brevipes* [35] and *Teucrium ramosissimum* [36]. In their study of hops, Kishimoto et al. [37] found that β-eudesmol promotes the spiciness of beer.

At the same time, β-eudesmol has many pharmacological activities. For example, it promotes intestinal motility and inhibits intestinal hyperactivity [38]. It can also counteract depression [39].

The results show that the four *Streptomyces* strains produced 31 terpenoids using five fermentation substrate hydrolysates during liquid fermentation, yielding a total terpene content of 989.94 μg/kg. When the fermentation substrate was all sorghum hydrolysate, the four strains of *Streptomyces* produced terpenoids up to a concentration of 521.56 μg/kg, yielding 14 terpenoids. When SPQ1 was used for fermentation, the concentration of terpenoids produced from the five raw materials was the highest, with a total of 16 terpenoids at 423.67 μg/kg.

By 28 days of solid-state fermentation, the four *Streptomyces* strains had produced 64 terpenoid species using five fermentation substrates, yielding a total terpenoid content of 23,651.52 μg/kg. When the fermentation substrate was sorghum, the four strains of *Streptomyces* produced the highest total concentration of terpenoids at 9815.65 μg/kg and 35 terpenoids in total. When SPS1 was used for fermentation, the concentration of terpenoids produced from the five types of raw materials was the highest at up to 15,605.74 μg/kg and 22 types of terpenoids.

During this study, we found that β-eudesmol was more prominent in the products of SPS1 and SPQ1, with both strains producing high concentrations of β-eudesmol. When the solid-state fermentation reached 28 days, SPS1 produced 4059.82 μg/kg of β-eudesmol when using a wheat solid medium and SPQ1 produced 1149.89 μg/kg of β-eudesmol when using a wheat medium.

In the liquid fermentation, although the concentration of β-eudesmol produced was considerably lower compared to that of the solid fermentation, β-eudesmol was still more prominent than other products of liquid fermentation. SPS1 and SPQ1 produced higher concentrations of β-eudesmol (146.54 μg/kg and 166.47 μg/kg, respectively) using a sorghum liquid fermentation medium. However, the production of β-eudesmol by the two strains decreased significantly when a wheat liquid fermentation medium was used; SPS1 did not even produce β-eudesmol. Thus, the mode of fermentation and the variation of the fermentation substrate have a large impact on the ability of the strain to produce β-eudesmol. This also confirms that the mode of fermentation, as well as the variation in the substrate, has an important influence on the mechanism of terpenoid formation in actinomycetes.

## 5. Conclusions

In this study, the terpenoid-producing ability of four strains of *Streptomyces*, namely SPQ1, SPS1, SPG1 and SPH1, was investigated. Five common Chinese traditional solid baijiu brewing materials were used as substrates, and four strains of *Streptomyces* were used for pure grain liquid fermentation and solid-state fermentation. GC-MS-HS-SPME was used to analyze the terpenoids in the product at the end of the fermentation.

The different fermentation substrates have an important influence on the terpenoids produced during the brewing process. Our study not only affirmed the status of sorghum as the main brewing raw material of nongxiangxing baijiu, but also proved that different materials can be used in the brewing of jiangxiangxing baijiu to provide a wider range of terpenoids and subsequent flavors for nongxiangxing baijiu. Different fermentation methods have a major impact on the way microorganisms grow and on the production of terpenoids. Exploring the effects of fermentation methods on terpenoid production by actinomycetes and the rational use of actinomycetes in the brewing process of baijiu will play an important role in the production of terpenoids.

## Figures and Tables

**Figure 1 foods-12-01494-f001:**
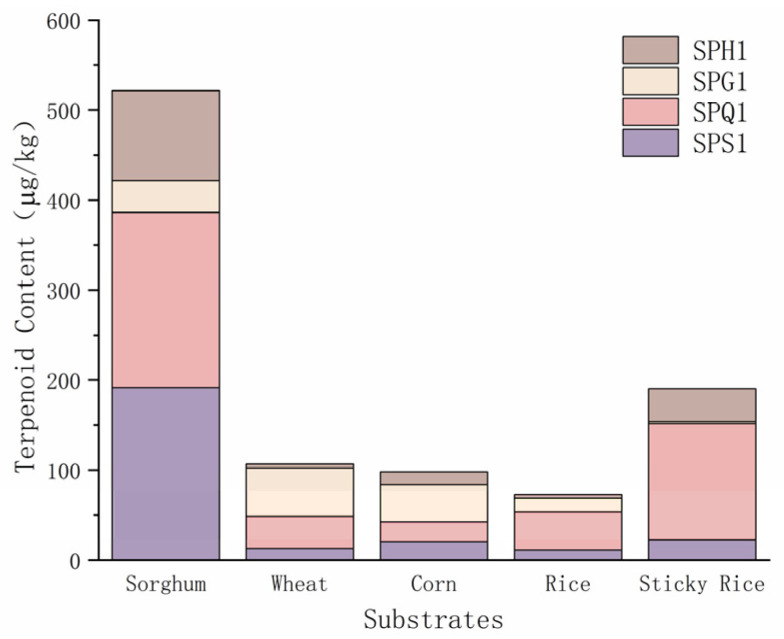
Content of terpenoid products from liquid-state fermentation of different types of substrates.

**Figure 2 foods-12-01494-f002:**
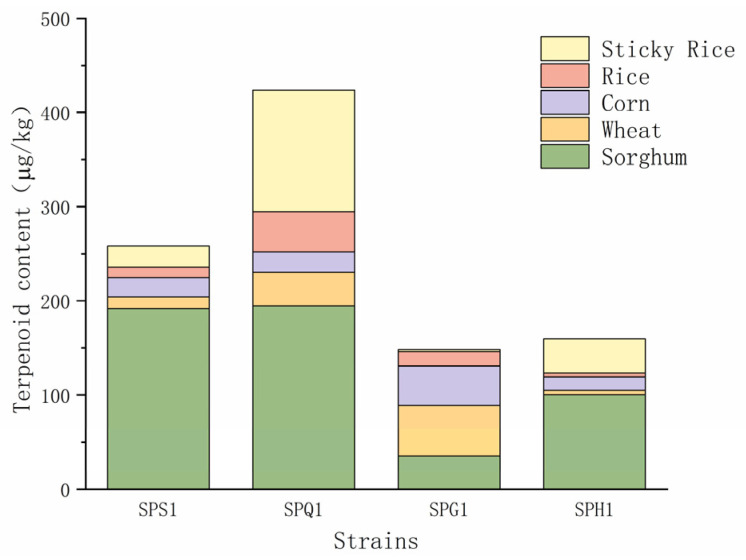
Content of terpenoid products from liquid-state fermentation of different strains.

**Figure 3 foods-12-01494-f003:**
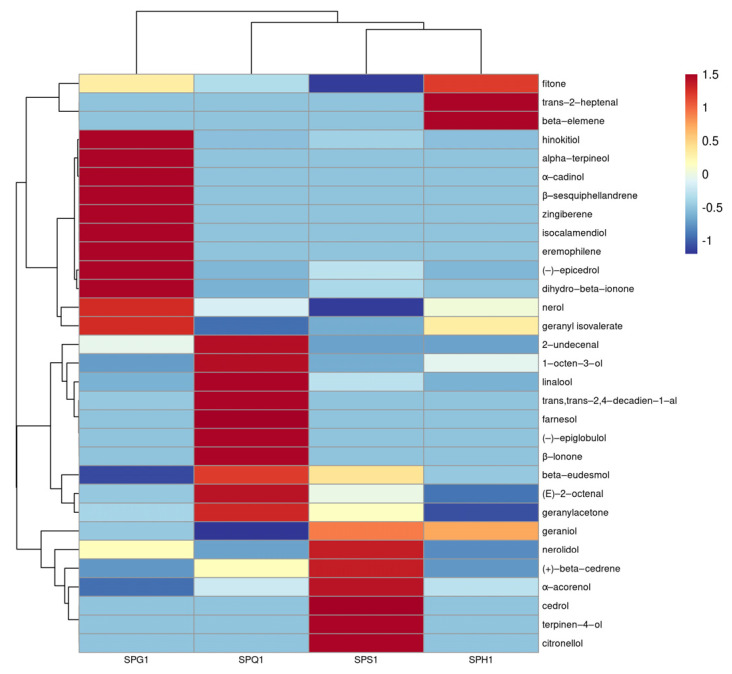
Heat map of the content of terpene products from liquid-state fermentation of different strains.

**Figure 4 foods-12-01494-f004:**
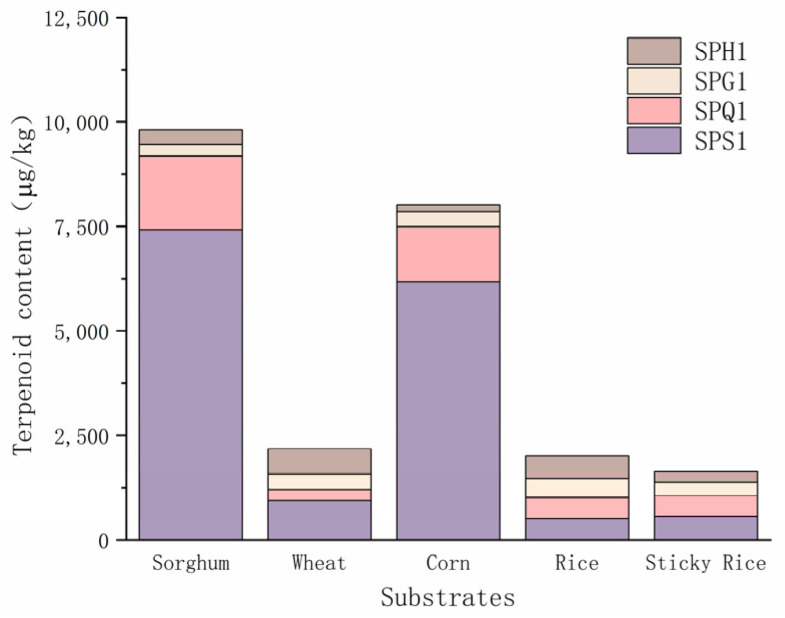
Content of terpenoid products from solid-state fermentation of different types of substrates at 28 days.

**Figure 5 foods-12-01494-f005:**
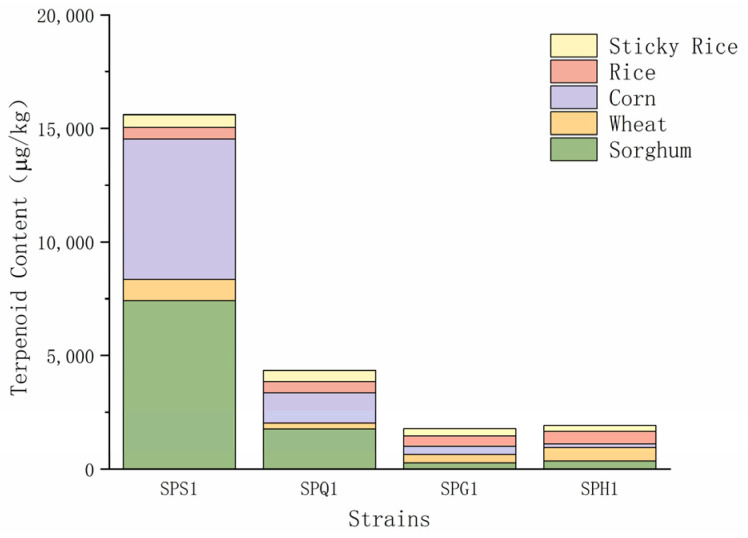
Content of terpenoid products from solid-state fermentation of different strains at 28 days.

**Figure 6 foods-12-01494-f006:**
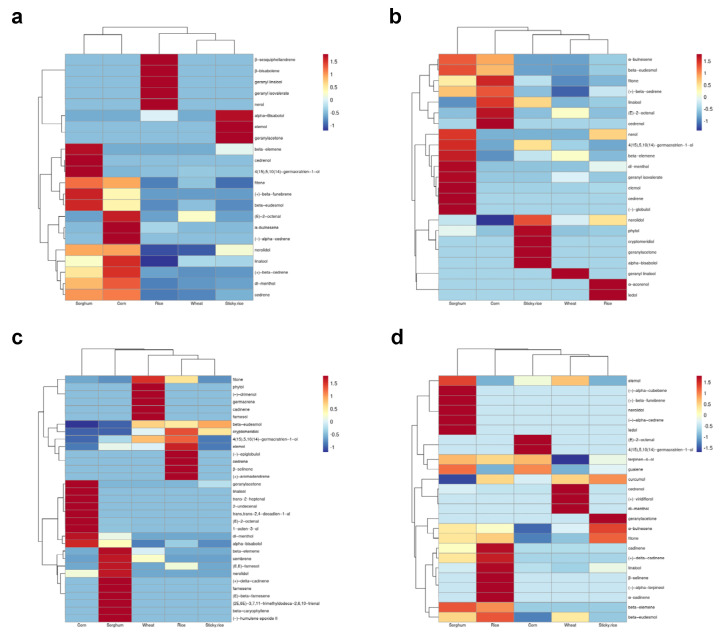
Heat map of terpene content in solid-state fermentation of SPS1, SPQ1, SPH1, SPG1 at 28 days. (**a**) Heat map of terpene content in solid-state fermentation of SPS1 at 28 days; (**b**) Heat map of terpene content in solid-state fermentation of SPQ1 at 28 days; (**c**) Heat map of terpene content in solid-state fermentation of SPG1 at 28 days; (**d**) Heat map of terpene content in solid-state fermentation of SPH1 at 28 days.

**Table 1 foods-12-01494-t001:** The content (μg/kg) of terpenoids produced by liquid fermentation ^a^.

	Terpenoids	Control ^b^	SPS1	SPQ1	SPG1	SPH1
Sorghum	1-Octen-3-ol	- ^c^	-	-	1.96	-
	(E)-2-Octenal	-	-	-	2.43	-
	Nerolidol	-	-	-	2.2	-
	Geranylacetone	-	-	-	3.44	-
	(+)-β-Cedrene	-	35.34	15.69	-	-
	β-Eudesmol	-	146.54	166.47	-	62.05
	Geranyl isovalerate	-	0.53	-	-	0.43
	Fitone	3.40	9.24	10.56	3.62	31.72
	(−)-Epiglobulol	-	-	1.92	-	-
	2,4-Decadienal	-	-	-	0.11	-
	2-Undecenal	-	-	-	5.97	-
	Zingiberene	-	-	-	6.37	-
	β-Sesquiphellandrene	-	-	-	9.04	-
	β-Elemene	-	-	-	-	5.93
Wheat	Hinokitiol	-	1.25	-	26.47	-
	Geraniol	-	1.68	-	-	-
	Linalool	-	1.74	0.94	-	1.55
	Nerolidol	-	5.75	-	-	-
	β-Eudesmol	-	-	23.33	-	-
	Geranyl isovalerate	-	-	-	3.93	-
	Fitone	1.73	1.82	5.84	22.34	2.02
	Cedrol	-	0.34	-	-	-
	Nerol	-	-	5.67	-	1.27
	(−)-Epicedrol	-	-	-	1.01	-
Corn	(E)-2-Octenal	-	3.62	-	4.42	-
	Nerolidol	1.79	5.22	3.28	4.34	2.86
	Geranylacetone	-	6.48	4.55	-	-
	Fitone	0.14	1.48	0.51	0.9	1.1
	Dihydro-β-ionone	1.45	1.69	1.50	3.3	1.58
	α-Acorenol	1.40	1.95	1.58	1.41	1.56
	2-Undecenal	-	-	7.16	-	-
	Nerol	-	-	-	13.04	5.23
	β-Ionone	-	-	3.16	-	-
	Geranyl isovalerate	-	-	-	0.1	1.84
	Isocalamenediol	-	-	-	0.14	-
	(−)-Epicedrol	-	-	-	0.33	-
	Eremophilene	-	-	-	2.06	-
	α-Cadinol	-	-	-	9.42	-
Rice	Nerol	-	0.35	-	-	0.16
	(E)-2-Octenal	-	7.42	-	-	2.87
	Geranylacetone	1.08	1.15	5.6	1.10	0.95
	Fitone	0.46	2.00	4.42	1.88	0.48
	Linalool	-	-	16.01	9.30	-
	β-Eudesmol	-	-	16.46	-	-
	(−)-α-Terpineol	-	-	-	3.13	-
Glutinousrice	(−)-Epicedrol	-	0.20	-	-	-
	Terpinen-4-ol	-	1.68	-	-	-
	Linalool	2.13	12.13	8.21	2.5	10.23
	Citronellol	-	2.55	-	-	-
	1-Octen-3-ol	-	3.28	46.7	-	15.52
	Geranylacetone	0.31	0.39	3.93	0.49	0.52
	Fitone	0.54	2.20	2.62	0.8	2.06
	2,4-Decadienal	-	-	10.8	-	-
	Geraniol	-	-	-	0.56	1.53
	2-Undecenal	-	-	12.05	-	-
	β-Eudesmol	-	-	18.02	-	-
	Farnesol	-	-	2.79	-	-
	(E)-2-Octenal	-	-	23.9	-	-
	(E)-2-Heptenal	-	-	-	-	6.68

^a^: Streptomyces was cultured at 180 rpm and 30 °C for 3 d; ^b^: Cereal extracts without inoculation; ^c^: - = not detected.

**Table 2 foods-12-01494-t002:** Content of terpenoids in blank raw material medium ^a^.

Terpenoids	Sorghum	Wheat	Corn	Rice	Sticky Rice
(E)-2-Octenal	- ^b^	-	0.63	-	-
Nerolidol	4.62	-	-	-	-
Fitone	4.69	8.17	10.46	2.66	3.01
Linalool	-	-	0.77	-	-
Geranylacetone	-	-	-	-	1.25

^a^: Cereal without inoculation; ^b^:- = not detected.

**Table 3 foods-12-01494-t003:** Content (μg/kg) of terpenoids produced from different substrates.

	7d ^a^					14d					21d					28d				
Compounds	1 ^b^	2	3	4	5	1	2	3	4	5	1	2	3	4	5	1	2	3	4	5
DL-Menthol	180.23	549.42	305.75	129.94	60.36	401.04	120.21	115.36	17.48	21.47	1294.79	116.77	805.01	90.71	33.92	401.75	98.5	507.95	17.39	42.53
Elemol	44.64	72.69	2.82	65.11	14.79	572.1	117.99	1.51	9.03	20.01	176.92	16.13	7.73	19.82	13.46	76.59	29.35	12.86	18.9	0.66
Linalool	80.36	154.97	57.4	27.59	9.00	23.75	51.88	20.13	111.51	15.31	60.96	51.37	70.89	69.69	11.41	28.85	23.51	38.07	59.05	35.39
(+)-β-Funebrene	60.13	- ^c^	44.89	-	0.55	11.5	-	38.98	-	0.09	887.77	-	304.84	-	-	447.77	-	208.95	-	-
Cedrenol	8.11	51.02	4.26	-	-	35.22	75.73	2.65	-	-	89.39	48.95	13.88	-	-	63.35	212.52	6.36	-	-
Cedrene	15.93	72.16	27.76	3.12	7.99	4.41	63.08	17.35	0.12	49.98	112.75	23.44	145.97	5.92	4.35	150.61	3.7	136.32	3.77	19.65
4(15),5,10(14)-Germacratrien-1-ol	481.55	443.89	24.11	53.26	63.64	305.22	230.05	7.66	20.58	126.01	1612.96	679.72	5.29	30.94	162.34	544.49	75.49	43.69	62.11	85.69
β-elemene	28.74	133.40	2.23	34.22	3.91	33.38	148.46	0.96	53.02	12.21	356.69	98.94	-	21.78	4.26	71.85	7.49	-	11.89	13.41
Nerolidol	167.55	270.25	42.12	89.16	76.89	178.32	15.33	25.88	14.02	17.34	221.01	22.02	115.57	32.46	19.12	79.49	21.64	67.72	24.24	64.12
Fitone	149.82	278.94	98.00	104.38	34.12	72.75	183.53	40.28	39.85	26.76	562.49	271.19	394.36	100.74	46.12	349.5	105.15	355.93	73.46	79.66
(+)-β-cedrene	185.68	778.62	111.95	208.62	47.82	71.43	123.77	290.62	32.97	59.11	5354.58	133.37	3806.73	204.23	98.33	1807.29	33.58	3064.30	214.09	103.31
β-eudesmol	1239.08	10276.62	816.81	2907.81	687.46	1831.98	5149.35	1057.8	1244.12	941.11	9635.02	2679.17	5496.13	1781.23	2118.22	5397.31	1356.9	3093.88	1083.94	796.40
Phytol	0.55	1.58	-	2.92	0.42	2.73	0.84	-	1.03	0.18	2.73	1.61	-	1.94	1.14	1.15	0.92	-	0.12	5.28
(E)-2-Octenal	58.38	8.84	99.85	0.11	1.68	105.67	4.31	41.21	0.03	0.46	4.40	1.31	84.56	-	-	0.2	2.22	42.16	-	-
Geranyl isovalerate	6.87	19.94	3.06	4.73	1.25	2.91	4.26	0.67	1.00	3.85	4.52	2.33	-	2.83	0.74	4.40	0.79	-	0.17	-
Nerol	1.47	20.79	-	5.86	-	0.81	3.14	-	0.39	-	6.66	-	-	6.82	-	9.33	-	-	6.60	-
α-Bulnesene	54.83	389.99	115.67	132.62	2.5	17.01	379.89	31.48	106.21	13.61	40.80	74.62	217.72	105.15	2.14	70.05	14.45	121.63	32.78	36.06
(−)-Globulol	11.48	-	-	-	-	3.98	-	-	-	-	48.57	-	-	-	-	62.98	-	-	-	-
α-Agarofuran	1.11	-	-	-	-	0.33	-	-	-	-	-	-	-	-	-	-	-	-	-	-
δ-Cadinene	39.89	112.58	-	12.31	-	97.69	69.42	-	63.57	-	119.74	8.09	-	43.52	-	20.19	-	-	34.16	-
2-Undecenal	11.66	-	52.07	-	-	0.15	-	22.33	-	-	-	-	35.12	-	-	-	-	8.37	-	-
β-Caryophyllene	2.45	-	-	-	-	54.60	-	-	-	-	36.14	-	-	-	-	1.79	-	-	-	-
(−)-Humulene epoxide II	5.84	-	-	-	-	107.95	-	-	-	-	109.68	-	-	-	-	5.97	-	-	-	-
Farnesal	160.07	33.26	14.26	-	-	1214.38	143.51	0.35	-	-	224.94	31.55	-	-	-	4.00	-	-	-	-
α-Farnesene	17.32	-	-	-	-	162.85	-	-	-	-	143.13	-	-	-	-	6.79	-	-	-	-
Cembrene	56.63	38.40	-	-	-	240.18	88.07	-	-	-	37.6	48.94	-	-	-	28.11	11.00	-	-	-
(E)-β-Famesene	48.83	-	-	-	-	276.67	-	-	-	-	72.34	-	-	-	-	4.64	-	-	-	-
α-Bisabolol	5.89	-	352.76	15.26	20.84	131.47	-	22.58	1.23	29.80	507.05	-	27.94	16.73	19.02	80.04	-	166.06	40.77	53.66
1-Octen-3-ol	111.44	14.35	532.74	-	6.76	1776.01	1118.52	124.24	-	1.20	1.68	2.04	227.3	-	-	0.15	-	42.41	-	-
trans-Farnesol	427.85	-	-	0.01	-	2872.72	-	-	0.89	-	923.86	-	-	11.00	-	51.08	-	-	17.91	-
α-Gurjenene	32.33	-	-	-	-	5.88	-	-	-	-	0.24	-	-	-	-	-	-	-	-	-
γ-Cadinene	14.06	38.33	-	4.48	3.14	4.52	111.99	-	27.02	6.25	14.45	12.19	-	17.90	3.06	2.2	0.29	-	7.99	0.13
α-Cedrene	1.94	-	1.04	-	-	1.21	-	1.19	-	-	3.36	-	11.11	-	-	2.3	-	13.7	-	-
Ledol	17.41	58.69	-	121.81	0.67	3.34	11.41	-	33.22	3.79	10.29	-	-	20.04	-	7.48	-	-	2.87	-
Guaiol	18.78	-	-	-	-	6.47	-	-	-	-	-	-	-	-	-	-	-	-	-	-
Terpinen-4-ol	0.02	2.69	0.56	0.41	0.67	0.74	2.76	0.96	3.57	0.61	1.02	3.48	1.42	3.61	0.57	2.39	1.41	2.40	2.31	2.04
α-Cubebene	0.27	-	-	-	-	18.38	-	-	-	-	14.35	-	-	-	-	13.06	-	-	-	-
β-Guaiene	7.17	7.25	7.1	46.17	0.90	1.37	2.21	3.07	12.28	2.00	3.65	-	5.10	0.18	1.33	8.43	-	7.83	-	2.75
α-Acorenol	-	46.55	-	10.43	1.32	-	17.34	-	1.61	0.38	-	0.85	-	8.28	-	-	-	-	1.38	-
Geranyl linallol	-	27.96	12.08	3.05	-	-	3.27	4.62	0.41	-	-	10.05	-	3.40	-	-	3.32	-	0.05	-
Cryptomeridiol	-	490.83	-	325.26	6.07	-	451.33	-	35.87	2.33	-	72.54	-	187.50	118.3	-	41.57	-	97.54	99.32
Farnesol	-	64.70	-	-	-	-	418.58	-	-	-	-	90.61	-	-	-	-	6.36	-	-	-
β-Germacrene	-	2.14	-	-	-	-	24.78	-	-	-	-	23.86	-	-	-	-	8.93	-	-	-
Drimenol	-	10.99	-	-	-	-	167.76	-	-	-	-	35.62	-	-	-	-	8.99	-	-	-
2,4-Decadienal	-	17.74	31.81	-	-	-	-	59.58	-	-	-	-	86.12	-	-	-	-	1.88	-	-
Curcumol	-	62.08	12.59	3.54	2.77	-	81.12	38.73	83.26	29.34	-	278.03	92.62	230.52	22.95	-	104.79	61.94	102.41	125.16
Viridiflorol	-	11.14	-	-	-	-	11.47	-	-	-	-	3.75	-	-	-	-	0.71	-	-	-
α-Pinene	-	28.60	-	-	-	-	14.38	-	-	-	-	6.16	-	-	-	-	-	-	-	-
Hinokitiol	-	-	3.53	-	-	-	-	1.09	-	-	-	-	-	-	-	-	-	-	-	-
Geranylacetone	-	-	5.10	-	22.59	-	-	1.63	-	11.40	-	-	3.83	-	22.70	-	-	2.26	-	76.44
7-epi-Sesquithujene	-	-	11.20	-	-	-	-	0.97	-	-	-	-	-	-	-	-	-	-	-	-
α-curcumene	-	-	22.31	-	-	-	-	0.02	-	-	-	-	-	-	-	-	-	-	-	-
(E)-2-Heptenal	-	-	16.51	-	-	-	-	28.22	-	-	-	-	37.35	-	-	-	-	15.61	-	-
Citronellol	-	-	0.74	-	-	-	-	0.61	-	-	-	-	-	-	-	-	-	-	-	-
2,6-Dimethyl-5-heptenal	-	-	2.68	-	-	-	-	0.9	-	-	-	-	-	-	-	-	-	-	-	-
(−)-Epiglobulol	-	-	14.68	7.42	-	-	-	4.01	0.12	-	-	-	-	4.64	-	-	-	-	4.80	-
β-Bisabolene	-	-	-	7.96	-	-	-	-	0.62	-	-	-	-	4.26	-	-	-	-	2.66	-
β-Sesquiphellandrene	-	-	-	7.85	-	-	-	-	2.09	-	-	-	-	17.66	-	-	-	-	10.78	-
β-Selinene	-	-	-	10.41	-	-	-	-	10.69	-	-	-	-	38.47	-	-	-	-	32.34	-
Aromandendrene	-	-	-	0.04	-	-	-	-	0.11	-	-	-	-	14.86	-	-	-	-	0.03	-
α-Terpieol	-	-	-	0.69	-	-	-	-	8.89	-	-	-	-	35.63	-	-	-	-	24.61	-
α-Cadinene	-	-	-	2.5	-	-	-	-	19.90	-	-	-	-	34.11	-	-	-	-	17.30	-
(E)-2-Octen-1-ol	-	-	-	-	0.52	-	-	-	-	0.07	-	-	-	-	-	-	-	-	-	-
(E)-2-Decenal	-	-	-	-	0.79	-	-	-	-	0.27	-	-	-	-	-	-	-	-	-	-

^a^: The fermentation condition was 30 °C for 28 days and samples were taken at 7 d intervals until the end of 28 d of fermentation; ^b^: 1 = sorghum, 2 = wheat, 3 = corn, 4 = rice, 5 = sticky rice; ^c^: - = not detected.

**Table 4 foods-12-01494-t004:** Content (μg/kg) of terpenoids produced from different strains.

	7d ^a^				14d				21d				28d			
Compounds	SPS1	SPQ1	SPG1	SPH1	SPS1	SPQ1	SPG1	SPH1	SPS1	SPQ1	SPG1	SPH1	SPS1	SPQ1	SPG1	SPH1
DL-Menthol	700.31	75.73	349.66	100.00	147.29	18.86	445.01	64.4	1438.62	113.04	768.60	20.94	836.15	51.86	110.49	69.62
elemol	77.06	66.33	25.75	30.91	29.34	8.71	662.22	20.37	13.46	51.85	147.04	21.71	0.66	36.26	31.18	70.26
linalool	53.35	28.33	19.58	228.06	13.51	11.02	1.70	196.35	89.38	15.78	6.31	152.85	18.67	12.96	2.53	150.71
(+)-β-Funebrene	67.94	16.35	7.91	13.37	34.01	12.81	0.94	2.81	1187.67	-	-	4.94	653.87	-	-	2.85
Cedrenol	4.84	4.26	- ^b^	54.29	28.90	2.65	-	82.05	85.50	13.88	-	52.84	47.99	6.36	-	227.88
Cedrene	57.96	9.76	13.31	45.93	67.01	5.72	52.92	9.29	257.70	13.44	21.29	-	288.99	21.29	3.77	-
4(15),5,10(14)-Germacratrien-1-ol	472.38	544.4	25.56	24.11	166.53	194.32	321.01	7.66	894.35	1086.38	505.23	5.29	449.34	207.60	110.84	43.69
β-elemene	20.62	78.64	9.20	94.04	4.15	39.55	101.27	103.06	111.34	93.38	227.40	49.55	47.69	21.49	12.38	23.08
nerolidol	251.76	257.74	125.31	11.16	38.50	37.51	174.53	0.35	299.02	36.79	69.46	4.91	203.46	45.58	5.41	2.76
fitone	271.44	185.17	75.90	132.75	96.78	101.57	107.48	57.34	680.14	539.82	90.11	64.83	597.40	280.87	22.89	62.54
(+)-β-cedrene	820.81	511.88	-	-	403.67	174.23	-	-	8686.98	910.26	-	-	4676.51	546.06	-	-
β-eudesmol	7012.8	7198.94	533.63	1182.41	2787.03	2660.94	3846.91	929.44	12288.20	6246.61	1649.91	1525.05	7560.39	2844.84	709.76	613.44
Phytol	-	3.89	1.58	-	-	3.94	0.84	-	-	5.81	1.61	-	-	6.55	0.92	-
(E)-2-Octenal	7.80	10.11	150.01	0.94	3.39	4.63	142.28	1.38	12.23	4.67	71.44	1.93	5.14	2.95	33.19	3.30
Geranyl isovalerate	11.52	16.51	1.05	6.77	3.67	5.75	0.50	2.77	2.90	7.38	-	0.14	0.17	5.19	-	-
Nerol	23.41	4.71	-	-	3.53	0.81	-	-	3.67	9.81	-	-	0.96	14.97	-	-
α-bulnesene	211.59	226.06	6.60	251.36	43.97	65.28	249.92	189.03	180.47	141.61	6.93	111.42	89.61	94.31	-	91.05
(−)-Globulol	-	11.48	-	-	-	3.98	-	-	-	48.57	-	-	-	62.98	-	-
α-Agarofuran	-	-	1.11	-	-	-	0.33	-	-	-	-	-	-	-	-	-
δ-Cadinene	-	-	2.90	161.88	-	-	87.46	143.22	-	-	94.52	76.83	-	-	5.49	48.86
2-Undecenal	-	-	63.73	-	-	-	22.48	-	-	-	35.12	-	-	-	8.37	-
β-Caryophyllene	-	-	2.45	-	-	-	54.60	-	-	-	36.14	-	-	-	1.79	-
(−)-Humulene epoxide II	-	-	5.84	-	-	-	107.95	-	-	-	109.68	-	-	-	5.97	-
Farnesal	-	-	207.59	-	-	-	1358.24	-	-	-	256.49	-	-	-	4.00	-
α-Farnesene	-	-	17.32	-	-	-	162.85	-	-	-	143.13	-	-	-	6.79	-
Cembrene	-	-	95.03	-	-	-	328.25	-	-	-	86.54	-	-	-	39.11	-
(E)-β-Famesene	-	-	48.83	-	-	-	276.67	-	-	-	72.34	-	-	-	4.64	-
α-Bisabolol	17.42	10.74	366.59	-	19.75	4.6	160.73	-	17.40	2.39	550.95	-	46.14	12.84	281.55	-
1-Octen-3-ol	-	1.77	663.52	-	-	0.47	3019.5	-	-	-	231.02	-	-	-	42.56	-
trans-Farnesol	-	-	427.86	-	-	-	2873.61	-	-	-	934.86	-	-	-	68.99	-
α-Gurjenene	-	-	-	32.33	-	-	-	5.88	-	-	-	0.24	-	-	-	-
γ-Cadinene	-	1.83	0.66	57.52	-	1.17	88.52	60.09	-	-	9.36	38.24	-	-	0.29	10.32
α-Cedrene	1.04	-	-	1.94	1.19	-	-	1.21	11.11	-	-	3.36	13.7	-	-	2.30
Ledol	-	121.73	-	76.85	-	13.10	-	38.66	-	8.32	-	22.01	-	2.87	-	7.48
Guaiol	-	-	-	18.78	-	-	-	6.47	-	-	-	-	-	-	-	-
Terpinen-4-ol	-	-	-	4.35	-	-	-	8.64	-	-	-	10.1	-	--	-	10.55
α-Cubebene	-	-	-	0.27	-	-	-	18.38	-	-	-	14.35	-	-	-	13.06
β-Guaiene	-	53.42	-	15.17	-	14.49	-	6.44	-	0.18	-	10.08	-	-	-	19.01
α-Acorenol	27.22	29.92	1.16	-	2.34	9.15	7.84	-	-	8.28	0.85	-	-	1.38	-	-
Geranyl linallol	7.40	35.69	-	-	3.04	5.26	-	-	3.40	10.05	-	-	0.05	3.32	-	-
Cryptomeridiol	63.00	617.28	10.59	131.29	0.67	80.31	394.59	13.96	-	39.28	337.72	1.34	-	35.64	202.79	-
Farnesol	-	-	64.70	-	-	-	418.58	-	-	-	90.61	-	-	-	6.36	-
β-Germacrene	-	-	2.14	-	-	-	24.78	-	-	-	23.86	-	-	-	8.93	-
Drimenol	-	-	10.99	-	-	-	167.76	-	-	-	35.62	-	-	-	8.99	-
2,4-Decadienal	-	-	49.55	-	-	-	59.58	-	-	-	86.12	-	-	-	1.88	-
Curcumol	-	-	-	80.98	-	-	-	232.45	-	-	-	624.12	-	-	-	394.30
Viridiflorol	-	-	-	11.14	-	-	-	11.47	-	-	-	3.75	-	--	-	0.71
α-Pinene	-	-	-	28.60	-	-	-	14.38	-	-	-	6.16	-	-	-	-
Hinokitiol	-	3.53	-	-	-	1.09	-	-	-	-	-	-	-	-	-	-
Geranylacetone	15.90	3.35	6.82	1.62	7.10	3.30	1.91	0.72	11.79	4.12	10.01	0.61	55.41	19.49	2.57	1.23
7-epi-sesquithujene	-	-	11.2	-	-	-	0.97	-	-	-	-	-	-	-	-	-
α-curcumene	-	-	22.31	-	-	-	0.02	-	-	-	-	-	-	-	-	-
(E)-2-Heptenal	-	-	16.51	-	-	-	28.22	-	-	-	37.35	-	-	-	15.61	-
Citronellol	-	-	0.74	-	-	-	0.61	-	-	-	-	-	-	-	-	-
2,6-Dimethyl-5-heptenal	-	-	2.68	-	-	-	0.90	-	-	-	-	-	-	-	-	-
(−)-Epiglobulol	-	-	7.42	14.68	-	-	0.12	4.01	-	-	4.64	-	-	-	4.80	-
β-Bisabolene	7.96	-	-	-	0.62	-	-	-	4.26	-	-	-	2.66	-	-	-
β-Sesquiphellandrene	7.85	-	-	-	2.09	-	-	-	17.66	-	-	-	10.78	-	-	-
β-selinene	-	-	8.40	2.01	-	-	0.64	10.05	-	-	21.72	16.75	-	-	17.38	14.96
Aromandendrene	-	-	0.04	-	-	-	0.11	-	-	-	14.86	-	-	-	0.03	-
α-Terpieol	-	-	-	0.69	-	-	-	8.89	-	-	-	35.63	-	-	-	24.61
α-Cadinene	-	-	-	2.50	-	-	-	19.90	-	-	-	34.11	-	-	-	17.3
(E)-2-Octen-1-ol	-	0.52	-	-	-	0.07	-	-	-	-	-	-	-	-	-	-
(E)-2-Decenal	-	0.79	-	-	-	0.27	-	-	-	-	-	-	-	-	-	-

^a^: The fermentation condition was 30 °C for 28 days and samples were taken at 7 d intervals until the end of 28 d of fermentation; ^b^: - = not detected.

## Data Availability

Data not available due to commercial restrictions.

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
