# Peer review of "Effects of Four Strains of Actinomycetes on the Content of Terpenoids in Baijiu"

_foods, 2023, doi:10.3390/foods12071494_

Round 1

Reviewer 2 Report

Dear authors, I have analyzed the manuscript and I would like to make some suggestions.

The abstract must be revised taking into account the criteria for writing an abstract for a scientific work: the numerical values must have the unit of measure but also to what it is related, g what? / kg what? ; all the methods used in the research must be listed; the last sentence must be removed because there are already published works that provide additional information on this subject (Research progress of trace components in sesame-aroma type of baijiu).

Please include in the paper the results of some statistical interpretations of the data. It's obligatory.

The Introduction should be improved with more information about this type of drink, in order to better understand the importance of the study.

Please remove the association between health benefits and an alcoholic drink.

Please rewrite sections 2.1 and 2.2, focusing on the differences between the two types of fermentation. Why is it called Solid-state fermentation if we have water in the mixture?

Please keep in mind that those who read the work do not all know how this product is made, therefore these sections must be very explicit, with quantities and percentages of each raw material that makes up a mixture.

The presented conclusions must be concise and very well structured, according to the requirements of the journal. That section should be reformulated; the information should be transferred to the Discussions section.

The paper must be reviewed by a native English speaker.

Round 2

Reviewer 2 Report

I agree with the publication in this form!

Author Response

亲爱的编辑和审稿人:

Thank you for your letter and for the reviewers’ comments concerning our manuscript entitled “Effects of four strains of Actinomycetes on the content of terpenoids in Baijiu” Those comments are all valuable and very helpful for revising and improving our paper, as well as the important guiding significance to our researches. We have studied comments carefully and have made corrections which we hope meet with approval. The following are the reviewers’ comments and our answer to their questions:

Reviewers' comments:

Reviewer #1:

1、I suggest to improve the statystical analysis. How did you estabish the significant differences? Plase provide more details.

Response: Thank you for your review of my article and your valuable comments.We have added Data analysis in the part of Materials and methods. When drawing a heat map of the terpene content in solid-state fermentation, the data was processed using the Lianchuan Biocloud platform, using Z-score method, and the data was clustered.

We tried our best to improve the manuscript and made some changes in the manuscript. These changes will not influence the content and framework of the paper.

We appreciate for Editors/Reviewers’ warm work earnestly and hope that the correction will meet with approval.

Once again, thank you very much for your comments and suggestions.

Sincerely yours,

Xuewu Guo
